# TANGO: Co-Speech Gesture Video Reenactment with Hierarchical Audio Motion Embedding and Diffusion Interpolation

**Haiyang Liu**[1]  **Xingchao Yang**[2]  **Tomoya Akiyama**[2]
**Yuantian Huang**[2]  **Qiaoge Li**  **Shigeru Kuriyama**[2,3]  **Takafumi Taketomi**[2]
[1]The University of Tokyo  [2]CyberAgent  [3]Toyohashi University of Technology

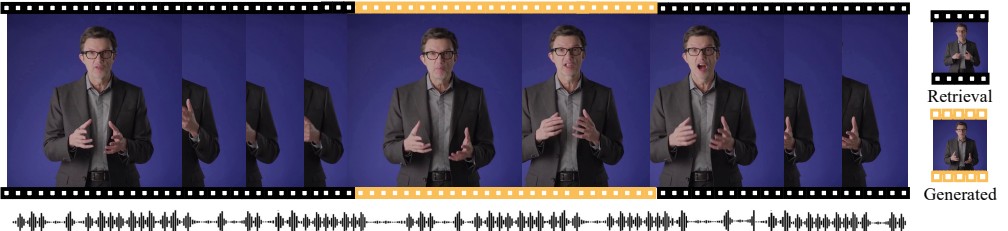

Figure 1: **TANGO** is a framework designed to generate co-speech body-gesture videos using a motion graph-based retrieval approach. It first retrieves most of the reference video clips that match the target speech audio by utilizing an implicit hierarchical audio-motion embedding space. Then, it adopts a diffusion-based interpolation network to generate the remaining transition frames and smooth the discontinuities at clip boundaries.

## Abstract

We present TANGO, a framework for generating co-speech body-gesture videos. Given a few-minute, single-speaker reference video and target speech audio, TANGO produces high-fidelity videos with synchronized body gestures. TANGO builds on Gesture Video Reenactment (GVR), which splits and retrieves video clips using a directed graph structure - representing video frames as nodes and valid transitions as edges. We address two key limitations of GVR: audio-motion misalignment and visual artifacts in GAN-generated transition frames. In particular, (i) we propose retrieving gestures using latent feature distance to improve cross-modal alignment. To ensure the latent features could effectively model the relationship between speech audio and gesture motion, we implement a hierarchical joint embedding space (AuMoCLIP); (ii) we introduce the diffusion-based model to generate high-quality transition frames. Our diffusion model, Appearance Consistent Interpolation (ACInterp), is built upon AnimateAnyone and includes a reference motion module and homography background flow to preserve appearance consistency between generated and reference videos. By integrating these components into the graph-based retrieval framework, TANGO reliably produces realistic, audio-synchronized videos and outperforms all existing generative and retrieval methods. Our code, pretrained models, and datasets are publicly available at https://github.com/CyberAgentAILab/TANGO.

## 1 Introduction

This paper addresses the problem of generating *high texture quality* co-speech body gesture videos from a reference speaker's talking video. Since significant progress has been made in talking face generation (Prajwal et al., 2020), our goal is to synchronize the body gestures in video with new, unseen speech audio. Successfully generating gesture-synchronized talking videos can significantly reduce production costs in real-world applications, such as news broadcasting and virtual YouTube content creation.

Generating gesture-synchronized videos from audio is promising but presents challenges, as humans are sensitive to both the video textural quality and the relationship between gestures and the audio's acoustic and semantic properties. Existing methods could be broadly categorized into two groups: *generative* and *retrieval*. Generative methods (Ginosar et al., 2019; Qian et al., 2021) generate all frames from given audio or audio estimated 2D pose using video generation neural networks (Chan et al., 2019), while retrieval methods (Zhou et al., 2022), recombine existing frames to match the audio and generate a few transition frames for recombination boundaries. Generative methods frequently suffer from artifacts such as temporal blur in hand and cloth textures. This limitation motivates our choice of the retrieval method, which ensures higher video quality for real-world applications.

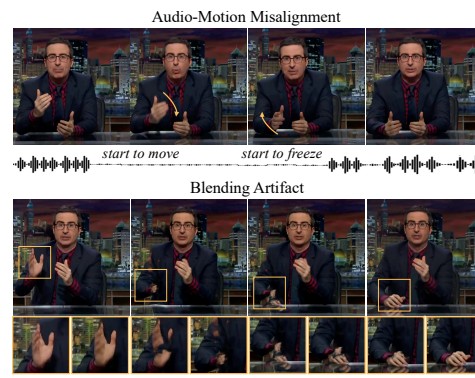

Figure 2: Limitations of GVR (Zhou et al., 2022).

Audio-Driven Gesture Video Reenactment (GVR) (Zhou et al., 2022), to the best of our knowledge, is the first and only retrieval-based method for gesture video generation. GVR splits videos into equal-length sub-clips and reassembles them in a motion graph-based (Kovar et al., 2008) approach. However, as shown in Figure 2, GVR has characteristic artifacts in two key components. First, the alignment between the target speech audio and the retrieved gesture video is limited, as the retrieval naively relies on audio onset features and keyword matching. Second, the performance of its GAN-based interpolation network is limited by its ability to predict accurate optical flow (Fleet & Weiss, 2006; Ilg et al., 2017), resulting in artifacts such as distorted hands.

To address these, we reproduce GVR's motion graph-based framework and introduce two improvements: an implicit feature distance-based gesture retrieval method and a diffusion-based interpolation network. The former (AuMoCLIP) introduces a hierarchical audio-motion joint embedding space to encode paired audio and motion modality data into a close latent space. The training pipeline is designed to split the low-level and high-level joint embedding space for learning local and global associations. After training, this joint embedding is adopted to retrieve gestures from the target's unseen audio. The latter, Appearance Consistent Interpolation (ACInterp), a diffusion-based interpolation network, leverages the power of existing video generation diffusion models, AnimateAnyone (Hu et al., 2023), to eliminate the blur and ghost artifacts found in traditional flow-based interpolation methods (Huang et al., 2022; Kong et al., 2022; Reda et al., 2022; Lu et al., 2022; Zhou et al., 2022), and proposes utilizing homography background flow and reference motion module to preserve appearance consistency between generated and reference videos. By integrating these improvements, our method, TANGO, could produce plausible videos while accurately aligning gestures with audio inputs. Our contributions can be summarized as follows:

- We propose a hierarchical audio-motion joint embedding space, AuMoCLIP, for accurately retrieving gestures based on target speech audio. To the best of our knowledge, AuMoCLIP is the first work to present CLIP-like embedding space between audio-motion modalities.

- We introduce a diffusion-based interpolation network, ACInterp, reducing spatial and temporal video artifacts and generating appearance-consistent video clips.

- We present a reproduced and improved motion graph-based gesture retrieval framework featuring a graph pruning method to generate co-speech gesture videos of infinite length.

- We release a small-scale, background-clean co-speech video dataset, YouTube Business, including data from 12 speakers to validate gesture video generation models.

- We integrate the above components into TANGO; it outperforms existing generative and retrieval methods, both quantitatively and qualitatively, on the existing Talkshow-Oliver and the newly introduced YouTube Business dataset.

## 2 RELATED WORK

Our methodology is related to prior research on generative and retrieval-based co-speech video generation, cross-modal retrieval, and video frame interpolation.

**Generative Co-Speech Video Generation.** Generative approaches (Qian et al., 2021; Liu et al., 2022a; Yoon et al., 2020; Liu et al., 2022b; Yang et al., 2023a;b; Zhu et al., 2023; Yi et al., 2023; Pang et al., 2023; Nyatsanga et al., 2023; He et al., 2024b) generate all frames from given audio via a two-stage pipeline. These methods, such as speech2gesture and speech-driven template (Ginosar et al., 2019; Qian et al., 2021), initially map audio to poses through specialized networks, followed by employing a separate GAN-based pose2video pre-trained model (Chan et al., 2019) to transform these poses into video frames. The audio2Pose stage has been improved by emotion-aware architecture (Qi et al., 2023) and diffusion models (Mughal et al., 2024). Recent literature has improved the performance of pose2video with diffusion models. For instance, AnimateAnyone (Hu et al., 2023) utilizes a reference-net attention-based motion module (Guo et al., 2023) for spatial and temporal consistency. Overall, the skeleton-level results from Generative methods are typically aligned with the audio, the pose2video stage (Chan et al., 2019; Zhang et al., 2023b; Hu et al., 2023; Rombach et al., 2022) is the bottleneck which often suffers from artifacts such as temporal blur in hands and cloth textures. This is due to the network's need to handle higher resolutions, long-term, accurate temporal consistency, and varied body deformations. This limitation shows the benefit of our approach, which reuses existing video frames. In our method, pose2video is only required for short clips with start and end frames; this allows to maintain high-quality results with minimal artifacts.

**Retrieval Co-Speech Video Generation.** Gesture Video Reenactment (GVR) (Zhou et al., 2022) represents the first attempt to retrieve gesture motion from speech audio using a motion graph-based (Kovar et al., 2008) framework. It has three key steps: (i) creating a motion graph based on 3D motion and 2D image domain distances, (ii) retrieval of the optimal path within this graph for the target speech by audio onset and keyword matching, (iii) blending the discontinues frames by an interpolation network based on flow warping and GAN. Our method improves GVR by incorporating learned feature-based retrieval and diffusion-based interpolation modules, resulting in better cross-modal alignment and high-quality transitions.

**Cross-Modal Retrieval.** Cross-modal retrieval aligns associations between different modalities within a learned feature space. The CLIP series (Radford et al., 2021; Li et al., 2022; 2023a) align text and images using contrastive learning. In the text-motion domain, MotionCLIP (Tevet et al., 2022) aligns motion with the frozen pretrained CLIP text space. GestureDiffCLIP (Ao et al.) aligns gesture motion with speech transcripts using max pooling. However, directly using text-only features to retrieve gesture motion is challenging due to the lack of timing information. Unlike previous methods, we propose a joint embedding space directly between audio and motion modalities.

**Video Frame Interpolation.** Video frame interpolation (VFI) aims to create intermediate frames between two existing frames. The integration of optical flow-based techniques with deep learning is the mainstream approach for VFI (Liu et al., 2017; Jiang et al., 2018; Niklaus & Liu, 2018; Xue et al., 2019; Niklaus & Liu, 2020; Park et al., 2020; 2021; Sim et al., 2021; Wu et al., 2022; Danier et al., 2022; Kong et al., 2022; Reda et al., 2022; Huang et al., 2022; Li et al., 2023b). Optical flow methods estimate pixel movement between frames to guide the interpolation process. However, these methods still face challenges such as handling high-frequency details, large or fast motions, occlusions, and balancing memory requirements. Hybrid models that combine CNNs with transformers (Lu et al., 2022; Zhang et al., 2023a; Park et al., 2023) and diffusion models (Voleti et al., 2022; Danier et al., 2024; Jain et al., 2024) have recently demonstrated more consistent and sharper results but require high memory capacity, e.g., 76GB training memory for 256x256 images (Lu et al., 2022; Danier et al., 2024), making the extension of these methods to real-world high-resolution videos challenging. Our method addresses these challenges by integrating latent diffusion-based architectures and temporal priority to improve visual fidelity and computational efficiency.

**Lip Synchronization in Co-Speech Video Generation.** Similar to previous co-speech gesture video generation works (Zhou et al., 2022; He et al., 2024a), our method only focuses on body gestures and employs post-processing using Wav2Lip (Prajwal et al., 2020) for lip synchronization. The reason for separating lip-sync and body gesture generation is performance-driven, *i.e.*, separated pipeline will have a higher SyncNet score (Prajwal et al., 2020). Audio correlates more strongly with lip movements than with body gestures, making it beneficial to handle them separately.

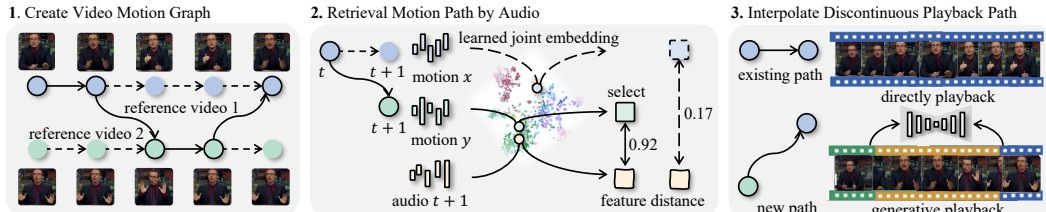

Figure 3: **System Pipeline of TANGO.** TANGO generates gesture video in three steps. Firstly, it creates a directed motion graph to represent video frames as nodes and valid transitions as edges. Each sampled path (in bold) dictates the selected playback order. Secondly, an audio-conditioned gesture retrieval module aims to minimize cross-modal feature distance to find a path where gestures best match target audio. Lastly, a diffusion-based interpolation model generates appearance-consistent connection frames when the transition edges do not exist in the original reference video.

# 3 TANGO

Our TANGO, as shown in Figure 3, is a motion graph-based framework for reenacting gesture videos based on target speech audio. Initially, we build upon the implementation of the VideoMotionGraph baseline (Zhou et al., 2022) and introduce graph pruning to create a directed Gesture Video Motion Graph (Section 3.1). In this graph, each node represents both the audio and image frames of the video, while each edge denotes a valid transition between frames. Given a target audio, its temporal features are extracted via a pre-trained audio-motion joint embedding network (AuMoCLIP). These features are then utilized to retrieve a subset of video playback paths (Section 3.2). When a transition edge does not exist within the original reference video, a frame interpolation network (ACInterp), is employed to ensure smooth transitions (Section 3.3). This enables each transi-

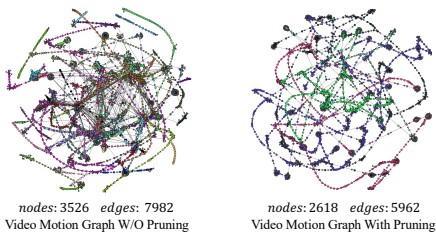

*nodes:* 3526  *edges:* 7982
Video Motion Graph W/O Pruning

*nodes:* 2618  *edges:* 5962
Video Motion Graph With Pruning

Figure 4: **Graph Pruning.** We delete paths with dead endpoints by merging SCC subgraphs. *i.e.*, those ending with a node without out-degree in the initial Gesture Video Graph (left), and obtain a strongly connected subgraph (right). Each node in the pruned graph is reachable from any other node within this subgraph, enabling efficient sampling of long video. The color of the paths represents different reference video clips for one speaker.

tion on the retrieved path to consist of realistic video frames. After the above three steps, TANGO reliably produces realistic, audio-synchronized gesture videos.

## 3.1 GRAPH CONSTRUCTION

**Graph initalization.** TANGO is represented as a graph structure $\mathbf{G}(\mathbf{N}, \mathbf{E})$ with nodes and edges. Similar to Gesture Video Reenactment (GVR) (Zhou et al., 2022), nodes $\mathbf{N} = \{\mathbf{n}_1, \mathbf{n}_2, \ldots, \mathbf{n}_i\}$ are defined as 1-frame, non-overlapping clips from reference videos, containing both RGB image frames and audio waveforms. The existence of valid transitions between nodes (edges)

---

**Algorithm 1** Graph Pruning Method for Enhancing Connectivity

Graph $G$ Enhanced Graph $G'$
Collect all SCC subgraphs in $G$ as $\mathbf{G}_{scc} = \{G_0, G_1, \ldots, G_n\}$ and $m = \mathrm{argmax}_k |G_k|$, where $|\cdot|$ denotes the size of a subgraph
**for** each subgraph $G_i (\neq G_m)$ in $\mathbf{G}_{scc}$ **do**
    **if** any nodes in $G_i$ not in $G_m$ **then**
        **for** each node $u$ in $G_m$ **do**
            **for** each node $v$ in $G_i$ **do** Calculate the distance $d(u, v)$ between nodes $u$ and $v$
$(i, j) = \arg \min_{u,v} d(u, v)$
Add bi-directional edges $\mathbf{e}_{i,j}$ and $\mathbf{e}_{j,i}$ to $G$

---

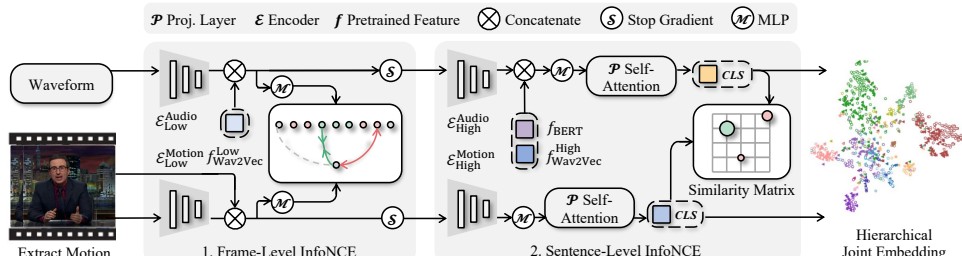

Figure 5: **AuMoCLIP.** AuMoCLIP is a pipeline to train hierarchical joint embedding. The audio waveform and extracted 3D motions are encoded in a learned embedding space where paired audio and motion have a closer distance than non-paired samples. It employs dual-tower encoder architecture; each encoder is split into low and high-level sub-encoder. Besides, it includes the pretrained Wav2Vec2 and BERT features to make it work. The embedding is trained with a frame-wise and clip-wise contrastive loss for local and global cross-modal alignment, respectively. We design the frame-wise loss by frames within a close temporal window ($i \pm t$) are positive, while distant frames ($i - kt$, $i - t$) and ($i + t$, $i + kt$) are negative.

$\mathbf{E} = \{\mathbf{e}_{1,1}, \mathbf{e}_{1,2}, \ldots, \mathbf{e}_{i,j}\}$ is determined on the basis of the similarities from both 3D motion space and 2D image space. We calculate the similarity in *3D space* from the positions of full body 3D joints. The 3D pose is extracted using a state-of-the-art open-source SMPL-X (Pavlakos et al., 2019) estimation method (Yi et al., 2023). The pose dissimilarity $\mathcal{D}_{\text{pose}}(\mathbf{n}_i, \mathbf{n}_j)$ between any pair of clips is determined by taking the average of the Euclidean distances for their positions and velocities across all joints.

The similarity in *2D image space* is the Intersection-over-Union (IoU) for body segmentation and hand boundary boxes. The body segmentation represents the visible foreground area in the image, computed by MM-Segmentation (Contributors, 2020), and the bounding boxes for hands are obtained from MediaPipe (Lugaresi et al., 2019). The ($1 - IoU$) of their visible surface areas then estimates the image space dissimilarity for each pair of frames as $\mathcal{D}_{\text{iou}}(\mathbf{n}_i, \mathbf{n}_j)$.

By employing the distance $d_{i,j} = \mathcal{D}_{\text{pose}}(\mathbf{n}_i, \mathbf{n}_j) + \mathcal{D}_{\text{iou}}(\mathbf{n}_i, \mathbf{n}_j)$, for any pair of nodes $\mathbf{n}_i, \mathbf{n}_j$, an edge $\mathbf{e}_{i,j}$ exists if their distances $d_{i,j}$ fall below predefined thresholds. We leverage an adaptive threshold by averaging the transition distances $t_{i,j} = (d_{i,i-1} + d_{i,i} + d_{i,i+1})/3$ in the original video.

**Graph Pruning.** The initial motion graph obtained from GVR is limited in connectivity, as shown in Figure 4, which reduces the efficiency of sampling a longer-length path. Search algorithms such as Beam search and dynamic programming typically encounter dead endpoints—nodes without outgoing edges—in the motion graph. In particular, the probability of sampling a path that ends at a dead endpoint increases with sample length, reaching 75.9% for randomly sampling a 10-second video and 98.6% for a 30-second video. To address this issue, we introduce a graph pruning method by merging the strongly connected component (SCC) subgraphs. A strongly connected component is a maximal subgraph where each node is reachable from any other node within the subgraph. Specifically, we employ Algorithm 1 to obtain a strongly connected component that can sample videos of any length starting from any point in the graph. More details for graph merging are in Appendix.

## 3.2 AUDIO-CONDITIONED GESTURE RETRIEVAL

The original Gesture Video Reenactment utilized onset and hard-coded keywords for audio-based path searching. However, this method has several limitations: i) speakers may not move synchronously with the audio onset; ii) the binary nature of onset results in weak distinction among similar samples; iii) there is no matching result if a keyword is not present in the reference video clips. These limitations lead to misaligned results. Therefore, we introduce a learning method to implicitly model the temporal association between audio and motion. As shown in Figure 5, our approach learns a hierarchical audio-motion joint embedding to consider short audio-motion beat alignment and longer-term content similarity simultaneously. To the best of our knowledge, our AuMoCLIP is the first pipeline to learn CLIP-like features between gesture audio and motion

modalities. We discuss our design in three key aspects: i) model architecture, ii) loss design, and iii) training schedule.

**Architecture of AuMoCLIP.** Inspired by the CLIP-based contrastive learning framework and Mo-CoV2 (Chen et al., 2020), we start from a dual-tower architecture trained with a global InfoNCE loss. Our key design for audio-motion modalities is the split between low-level and high-level encoders. Following Wav2Vec2 (Baevski et al., 2020), we represent audio as a raw waveform and use a 7-layer CNN (low-level) and a 1-layer Transformer (high-level) for the audio encoder. For motion, inspired by NeMF (Guo et al., 2022), we use a 15D representation and a motion encoder consisting of a 28-layer CNN (modified from TM2T (Guo et al., 2022)) and a 1-layer Transformer. The first 10 layers of the CNN are used as the low-level motion encoder. As shown in Figure 5, we use a Projection MLP to map low-level features, while a Projection Self-Attention to obtain the CLS token to summarize high-level features.

Since Wav2Vec2 is trained on large-scale human speech audio, we concatenate the frozen pretrained low-level and high-level features from Wav2Vec2 to enhance performance. However, encoding only audio waveforms is insufficient for high-level mapping between speech audio and gesture motion because gestures are often related to speech transcripts, while Wav2Vec2 and our audio encoder focus on "audio texture." To address this, we include timing-aligned BERT features using the Wav2Vec2CTC model and pretrained BERT. We design a word timing alignment method to align BERT features correctly without relying on MFA (see Appendix for details). This allows the audio branch to contain the necessary features for training the joint embedding.

**Local and Global Contrastive Loss.** We retain the InfoNCE loss for the CLS token in the mini-batch as the global contrastive loss and introduce a local contrastive learning task. Specifically, as shown in Figure 5, we define the frame-wise loss where frames within a close temporal window $(i \pm t)$ are considered positive, while distant frames $(i - kt, i - t, i + t, i + kt)$ are considered negative. In this paper, we set $t = 4$ and $k = 4$ for 30 FPS motion. This design proposes an easier learning task by accounting for slight misalignments in natural talking scenarios.

**Stop Gradient for Low-Level Encoders.** We aim to maximize both low-level and high-level retrieval accuracy during training. Our observation shows that directly optimizing both losses decreases the performance of the low-level encoder but improves the high-level encoder. This suggests that: i) the low-level encoder should not be trained jointly with the high-level encoder, and ii) including low-level features benefits the high-level encoder. Therefore, we stop the gradient from the global contrastive loss to the low-level encoder. This operation enables us to maximize the performance of both feature sets.

**Feature-Based Gesture Retrieval.** After training, we obtain two types of features: low-level features that can distinguish whether the current 8-frame audio-motion pair is matched, and high-level features that can evaluate if the current 4-second audio-motion clip is paired. We leverage these features for retrieval by combine these two features, First, for each 4-second clip groundtruth motion, we pre-calculate the high-level feature and repeat it for each frames. Next, we pre-calculate low-level feature and directly use its per-frame feature value. We then search for the best-matched path $P_{both\_match}$ by maximizing the both low-level and high-level similarity between the motion and the target audio over the entire path via Dynamic Programming (DP).

### 3.3 DIFFUSION-BASED VIDEO FRAME INTERPOLATION

The transition frames, synthesized from previous flow-based methods, often suffer from blur artifacts. To improve this, as shown in Figure 6, we leverage the power of the two-stage (pose2image and image2video) video generation diffusion model, AnimateAnyone (Hu et al., 2023). Our method, ACInterp, generates target interpolation frames $t \in (i, j)$ using the existing start frames $t \in (i-k, i]$ and end frames $t \in [j, j + k)$, along with linear-blended 2D pose images and homography background offsets.

**Homography Offset-Refined Pose2Image Stage.** As shown in Figure 6, The Pose2Image stage aims to sample a random noise $z_t$ and denoises it for estimated image latent $\hat{z}_0$. As same as AnimateAnyone(Hu et al., 2023), ACInterp i) implements denosing progress in a latent space with pretrained VAE Encoder and Decoder $\mathcal{E}_{VAE}, \mathcal{D}_{VAE}$; ii) adds pose features from PoseGuider $\mathcal{G}$ to noisy latent space as the input to DenoisingNet $\mathcal{D}$; iii) incorporates a ReferenceNet $\mathcal{R}$ and CLIP Im-

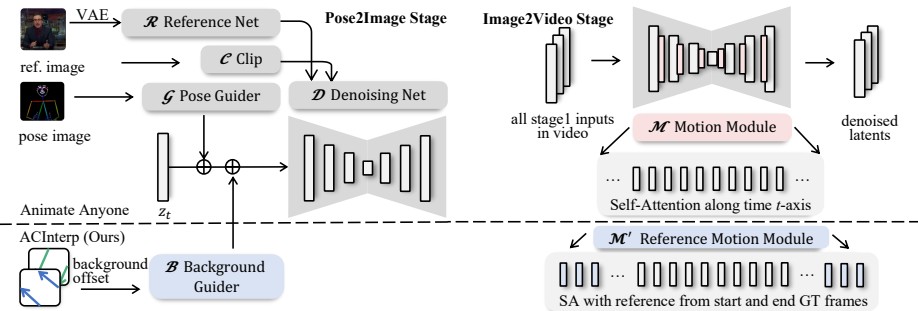

Figure 6: **ACInterp.** Our Appearance Consistent Interpolation (ACInterp) model generates target interpolation frames $t \in (i, j)$ using the existing start frames $t \in (i - k, i]$ and end frames $t \in [j, j + k)$, along with linear-blended 2D pose images and homography background offsets. Based on AnimateAnyone, ACInterp enhances appearance consistency in two ways. First, during the pose-to-image stage, estimated background pixel offsets are introduced to produce background-stable image results. Second, it uses the start and end frames as temporal priorities for the Motion Module to ensure human identity consistency. Achieving appearance consistency in transition frames is crucial for making Gesture Video Graph results appear natural.

age Encoder $\mathcal{C}$ to preserve consistent objects' appearances. Hierarchical features from the merged reference latent are concatenated to corresponding layers of the target DenoisingNet $\mathcal{D}$ to embed identity information. iv) calculates v-prediction (Rombach et al., 2022) loss for training.

Different from AnimateAnyone, we introduce additional homography background offset flow to eliminate artifacts due to camera parameter changes. As shown in Figure 7, the generated images often cause the drift of objects in a background, ignoring their fixed position in the reference background image. This is caused by overfitting the camera parameter changes in in-the-wild videos. To address this, we calculate an image-level background offset flow $H_{i,k} \in \mathbb{R}^{h \times w \times 2}$ and add it by BackgroundGuider $\mathcal{B}$, the $\mathcal{B}$ has the same architecture with $\mathcal{G}$. Specifically, we compute the homography matrix between the reference and target images to calculate the pixel movement $\Delta x, \Delta y$ for the background region. In particular, we masked the foreground by human segmentation results from DeepLabv3, then applied SIFT, FLANN, and RANSAC to keypoint detection, keypoint matching, and homography matrix computation, respectively.

Figure 7: **Appearance Artifacts** in AnimateAnyone (Hu et al., 2023). The generated results (in the second frame) show i) position jitters due to training data camera movement, and ii) identity inconsistencies after the Motion Module refinement. *Best viewed with Acrobat Reader. Click the images to play the clip.*

**Reference Motion Module-based Image2Video Stage.** As shown in Figure 6, the Image2Video stage captures temporal dependencies among video frames to mitigate the jitter effects in the Pose2Image stage. AnimateAnyone (Hu et al., 2023), as same as AnimateDiff (Guo et al., 2023), optimizes a residual self-attention-based motion module $\mathcal{M}$ within DenoisingNet $\mathcal{D}$ in this stage. The motion module reshapes a feature map $x \in \mathbb{R}^{b \times t \times h \times w \times c}$ to $x \in \mathbb{R}^{(b \times h \times w) \times t \times c}$ and then performs temporal attention along the $t$ dimension. However, as illustrated in Figure 7, this approach tends to produce averaged appearances across image sequences, resulting in diminished identity consistency. While this artifact may be negligible for other tasks where human identity is not important, it substantially degrades the realism of our gesture video graph.

Our analysis identifies that the key issue is the expansive solution space for the motion module, as self-attention is applied exclusively along the temporal dimension $t$. To address this, we introduce additional conditioning to constrain the solution space. We train the motion module by randomly selecting start and end pixels to effectively reduce uncertainty. During training, for the feature map $x \in \mathbb{R}^{(b \times h \times w) \times t \times c}$, we introduce a probability $p$ to incorporate 4-frame ground truth latent

Table 1: Evaluation for co-speech video generation on Show-Oliver and YouTube Video dataset.

| | Show-Oliver | | | | YouTube Talking Video | | | |
| | FVD ↓ | FGD ↓ | BC ↑ | Diversity ↑ | FVD ↓ | FGD ↓ | BC ↑ | Diversity ↑ |
|---|---|---|---|---|---|---|---|---|
| Ground Truth | - | - | 0.326 | 3.514 | - | - | 0.435 | 3.746 |
| SpeechDrivenTemplate (Qian et al., 2021) | 2.239 | 5.722 | **0.401** | 1.950 | 7.612 | 5.559 | 0.461 | 2.081 |
| ANGIE (Liu et al., 2022c) | 2.079 | 5.112 | 0.359 | 2.577 | - | - | - | - |
| S2G-Diffusion (He et al., 2024a) | 2.007 | 4.799 | 0.393 | 3.398 | 5.835 | 5.011 | 0.439 | 2.625 |
| GVR (Zhou et al., 2022) | 1.615 | 4.246 | 0.270 | 4.623 | 4.027 | 2.900 | 0.331 | 3.573 |
| TANGO (Ours) | **1.379** | **3.714** | 0.375 | **5.393** | **3.133** | **2.068** | **0.479** | **4.128** |

features as *reference*. Leveraging classifier-free guidance, our conditional diffusion-based motion module supports inference with or without these reference frames. The 4-frame segment alignment corresponds to the node length in the Gesture Video Reenactment. Finally, During inference, we introduce $4 \times \alpha$ and $4 \times \beta$ start and end conditional frames, generating intermediate 8 frames for transition edges.

# 4 EXPERIMENTS

## 4.1 DATASET

Our experiments are conducted on the open-source Show dataset (Yi et al., 2023) and a newly collected YouTube Video dataset. The Show dataset comprises 26 hours of talking videos featuring four speakers with varying backgrounds and irregular camera movements. We selected the speaker, Oliver, as these videos contain fewer interactions with the background. The YouTube Video data is a small-scale, less than one hour dataset from in-the-wild YouTube videos characterized by clean backgrounds and fixed camera positions. More details of split for each dataset is in the APPENDIX.

## 4.2 EVALUATION OF GENERATED VIDEOS

We compare our method with the previous state-of-the-art Generative method SDT (Qian et al., 2021), ANGIE (Liu et al., 2022c), S2G-Diffusion (He et al., 2024a) and reassemble-based method GVR (Zhou et al., 2022). We use the pertained weights from SDT and finetune the pose2img stage with a specific cloth. We evaluate ANGIE with the original paper test samples provided on Show-Oliver. For GVR, we reproduce the onset-based graph search and pose-aware neural rendering according to the implementation details in their paper.

**Objective Evaluation** We employ both video and kinematic Feature Distance (FVD (Carreira & Zisserman, 2017) and FGD (Yoon et al., 2020)) to quantify feature-level discrepancies. Additionally, we utilize Beat Consistency (BC)

Table 2: User Study on Talkshow-Oliver.

| | SDT | GVR | TANGO (Ours) | GT |
|---|---|---|---|---|
| Video Texture Quality ↑ | 4.1% | 26.7% | **33.8%** | 35.5% |
| Audio-Motion Alignment ↑ | 29.2% | 10.9% | **28.7%** | 31.2% |
| Overall Preference ↑ | 4.9% | 20.6% | **36.9%** | 37.6% |

(Liu et al., 2022d) and Diversity (Li et al., 2021) metrics to evaluate audio-motion synchronization and gesture diversity, respectively. As shown in Table 1, our method outperforms GVR and SDT across all metrics except for BC. The fully generated baseline exhibits greater flexibility in the output motion space, which results in better BC performance. However, compared to SDT, our method significantly improves video quality. Furthermore, compared to GVR, our approach consistently shows improvement across all metrics, demonstrating that TANGO generates more realistic and audio-synchronized videos.

**Subjective Evaluation** As shown in Table 2, we conducted a user study across four results. 47 Participants were asked to assess each video based on i) which video is more physically accurate, ii) which video's content aligns more closely with the audio, and iii) overall, which video is more like a real video. Users compared videos from all four results in a single row, with the order of videos randomly shuffled. A total of 60 video clips, each spanning 6 sec. We didn't include ANGIE in the user study due to not having enough result video clips. Some snapshots of a transition period are shown in Figure 8, and the supplementary material includes video results. Our method scored comparable to the ground truth video and outperforms the existing generative method SDT and retrieval method GVR with a clear margin.

## 4.3 EVALUATION OF AUMOCLIP

We evaluate the effectiveness of AuMoCLIP by retrieving gesture motion sequences using target audio features and measure performance using retrieval accuracy. As shown in Table 3, we compute low-level and high-level retrieval accuracy. Low-level retrieval accuracy is calculated by randomly selecting an audio feature at frame $i$ and finding the motion frame within $(i - 16, i + 16)$ with the highest cosine similarity. If this frame lies within $(i - 4, i + 4)$, it is marked as accurate. The final accuracy is averaged over 16K random pairs, so the random search will have an accuracy of 25%. Similarly, high-level retrieval accuracy is measured by selecting an audio high-level feature and comparing it against 256 motion candidates (1 paired motion + 255 negatives). If the highest cosine similarity corresponds to the paired motion, it is marked as accurate. This accuracy is averaged over 3K random pairs, and the random search will perform 0.391%. If the performance beats the random search, that means the model works correctly. The onset and keyword matching in GVR has a performance of 35.38% (low-level) and 1.288% (high-level), which is better than the random search. We then discuss the performance roadmap to the final AuMoCLIP.

**Generative Features.** One straightforward approach is to directly train an audio-to-motion network (Liu et al., 2022b) and compute the joint level distance between the generated motion and motion candidates for retrieval. However, this method yields lower-than-expected performance, achieving only 29.03% on low-level retrieval and 1.403% on high-level retrieval.

**MaxPooling or CLS Token.** We then switch to a MoCoV2-based dual-tower contrastive learning framework, starting with high-level features only. MoCoV2 is originally designed for images, not sequential data like motion. One solution is to apply max pooling along the time axis for the global token, similar to (Ao et al.). However, while max pooling focuses on accurate local alignment, it is less effective for global retrieval, resulting in a performance of only 5.312%. To address this, we adopt an adaptive global feature merge approach using the CLS token, which significantly improves performance to 11.84%. We keep CLS token in the remained experiments.

Table 3: Comparison of features for audio-motion retrieval

|  | Low Level | High Level |
|---|---|---|
| Random Search | 25.00% +00.00% | 0.391% +00.00% |
| Generative Features | 29.03% +16.10% | 1.403% + 258.8% |
| Keyword (Zhou et al., 2022) | - | 1.288% + 229.4% |
| Onset (Zhou et al., 2022) | 35.38% +41.51% | - |
| Baseline (Max Pooling) | - | 5.312% + 1360% |
| Baseline (CLS Token) | - | 11.84% + 3028% |
| + Wav2Vec2 | - | 12.73% + 3255% |
| + BERT | - | 15.68% + 4010% |
| + Wav2Vec2&BERT | - | 16.40% + 4194% |
| + Split (Low + High) | 47.94% + 99.76% | 17.83% +4460% |
| + Split (Low only) | 65.57% + 162.2% | - |
| AuMoCLIP (+ Stop Grad.) | **65.68% + 163.8%** | **19.54% + 4897%** |

**Pretrained Audio Features from Wav2Vec2 and BERT.** As shown in Table 3, incorporating pretrained audio features, specifically time-aligned BERT features, significantly enhances the baseline's performance from 11.84% to 15.68%. This improvement occurs because BERT captures high-dimensional language semantics rather than just "audio textures.", which is critical for co-speech gesture retrieval task.

**Discussion of Low-Level Contrastive Learning.** Including the low-level contrastive learning task consistently benefits high-level retrieval performance, suggesting that adding a more robust low-level feature improves high-level performance. Interestingly, we found that training only the low-level contrastive learning task achieves significantly better performance, reaching 65.57%. These observations suggest that we should: i) incorporate learned low-level features into high-level embedding learning, and ii) avoid the influence of high-level learning on low-level features. Therefore, we propose simply stopping the gradient to achieve the best performance for both features.

## 4.4 EVALUATION OF VIDEO BLENDING METHODS

We compare the effectiveness of the proposed diffusion-based video frame interpolation method by evaluating the quality of blended videos. The test set is from the same videos as other sections. Since the test videos in other sections vary in length (e.g., 3 to 10 seconds). In this section, we evenly sampled 8-frame clips, resulting in a 368-video test set for evaluating blending. Our approach is compared with the Pose Aware Neural Rendering in the original GVR (Zhou et al., 2022), the state-of-the-

Table 4: Comparison of video blending methods.

|  | PSNR ↑ | LPIPS ↓ | MOVIE ↓ | FVD ↓ |
|---|---|---|---|---|
| FiLM (Reda et al., 2022) | 35.43 | 0.072 | 74.85 | 1.358 |
| VFIF (Lu et al., 2022) | 34.91 | 0.077 | 79.42 | 1.777 |
| AnimateAnyone (Hu et al., 2023) | 32.63 | 0.127 | 86.06 | 1.421 |
| PANR (Zhou et al., 2022) | 35.18 | 0.071 | 75.02 | 1.190 |
| ACInterp (Ours) | **35.63** | **0.065** | **72.65** | **0.922** |

| Animate Anyone | FiLM | VFIFormer | Pose Aware Neural Rendering | Ours | Ground Truth |

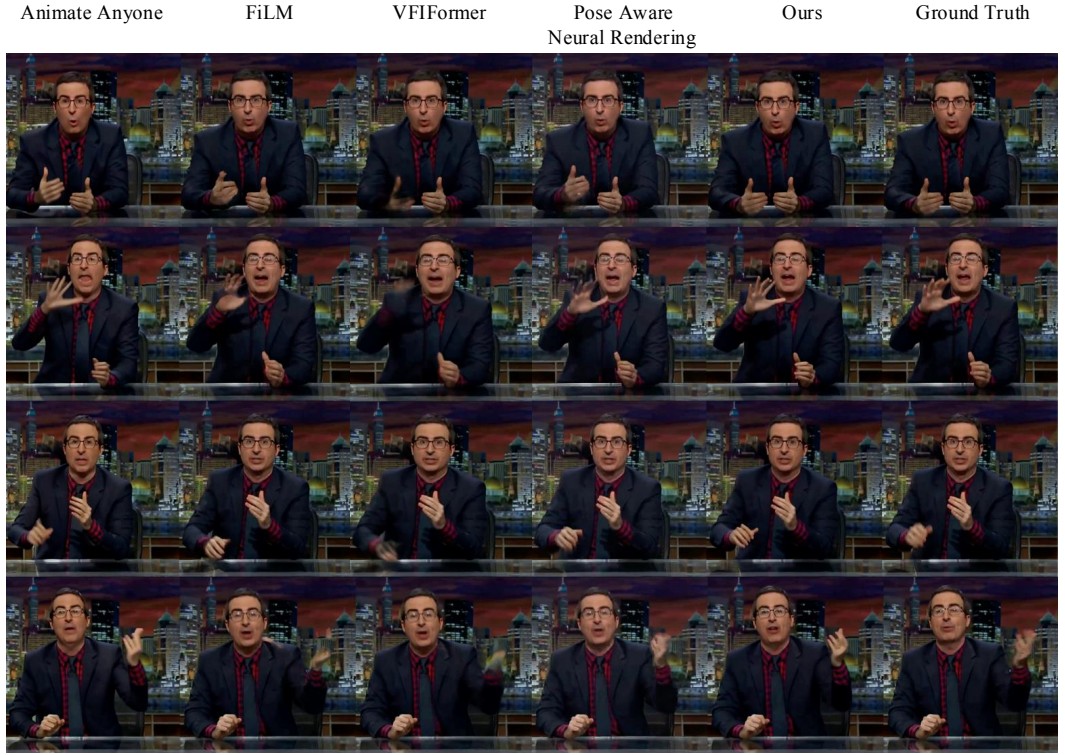

Figure 8: **Comparison for Transition Frames Generation.** From top to bottom, we show four snapshots in the same frames across different methods. Our method shows fewer artifacts in the hand regions and maintains appearance consistency with the GT frames. AnimateAnyone can recover the hands but loses appearance consistency. The flow-based methods FiLM and VFIFormer fail to estimate the flow for complex motions, resulting in the disappearance of hands. Pose Aware Neural Rendering shows better hand results but still suffers from artifacts such as blurring.

art flow-based blending method FiLM (Reda et al., 2022), VFIFormer (Lu et al., 2022), and the diffusion-based pose-guided video generation method AnimateAnyone (Hu et al., 2023). All methods are re-trained on the Show-Oliver dataset with a training and inference resolution of $768 \times 768$. We evaluate both image-level and video-level quality. For single images, we utilize Image Error (L1 pixel-level distance), Learned Perceptual Image Patch Similarity (LPIPS), and Peak Signal-to-Noise Ratio (PSNR). For videos, we adopt the Mean Opinion Video Quality Estimation (MOVIE) and feature-level distance (FVD). The features for calculating LPIPS and FVD are obtained from pretrained AlexNet and I3D networks, respectively. The objective comparisons are shown in Table 4. Overall, our diffusion-based interpolation model outperforms both previous flow-based and diffusion-based methods by a clear margin, *e.g.*, FVD 0.922 vs. the previous best 1.190. See Figure 8 and supplementary material for the resulting video.

## 5 CONCLUSION

We presented TANGO, a framework for generating high-fidelity videos where body gestures align with the target speech audio. To the best of our knowledge, TANGO is the first work to present CLIP-Like contrastive learning on audio and motion modalities, and it is the first open-source motion graph and audio-driven video generation pipeline. In the future, we aim to extend the gesture video graph on general human motion videos, such as dance, sports and more.

**Disclosures.** The authors acknowledge that they are solely responsible for ensuring the legal compliance of their collected benchmark data. Specifically. **Show-Oliver** is derived from the publicly available TalkShow dataset (Yi et al., 2023). To ensure compliance with YouTube's terms of service, we provide the original video URLs along with the data preprocessing scripts for **YouTube Business Dataset**, allowing researchers to reconstruct the dataset under the same conditions.

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

# A   APPENDIX

## A.1   DATASETS

**Show-Oliver.** The Show dataset comprises 26 hours of talking videos featuring four speakers with varying backgrounds and irregular camera movements. We selected the speaker, Oliver, as these videos contain fewer interactions with the background. The full Show-Oliver dataset contains 6546 video clips, ranging from 3 to 10 seconds. We evaluate our approach using multiple few-shot sets, each derived from a total of 10 minutes of randomly selected video clips with consistent clothing. These sets are divided into 80%, 10%, and 10% splits for training, validation, and testing, respectively.

**YouTube Video Dataset.** We further collect and process a small-scale, few-shot dataset from in-the-wild YouTube videos characterized by clean backgrounds and fixed camera positions. These videos feature 12 speakers delivering presentations lasting 1 to 2 minutes. We select four speakers of them and picks 6 to 10-second video subsets as the test set and use the remaining videos for training and constructing the Gesture Video Reenactment.

We first collect raw YouTube videos featuring 12 different identities. These videos are then segmented into multiple clips based on the detected sentence boundaries in the audio. Face detector is utilized to ensure that all clips contain clear faces. Subsequently, post-processing is employed to eliminate background obstructions and automatically adjust the camera position for consistency. Finally, we obtained 304 clips for different identities, with from total duration of 1 hour data. For each identity, the longer video clips are used for the validation set, while the remaining clips constitute the training set.

## A.2   TIMING-ALIGNED BERT FEATURES WITHOUT MFA

To achieve time-aligned BERT features for audio without forced alignment (MFA), we combine Wav2Vec2 and BERT models through the following steps:

**Transcription (ASR).** We use Wav2Vec2 with a Connectionist Temporal Classification (CTC) head to obtain the logits, which represent the model's confidence scores for each possible token at each time step. The logits are processed to generate a sequence of predicted token IDs for the audio input. These token IDs are then decoded into a transcription using the Wav2Vec2 processor's vocabulary. For example, we have the alphabet sequence $["", "", "T", "", "", "h", "e", "", "F", "i", "r", "s", "t"]$ in this step.

**BERT Embedding.** The transcription is tokenized using the BERT tokenizer, and the embeddings are obtained from the BERT model. The tokenizer in BERT conver the alphabet sequence into word sequence $["CLS", "The", "First", "POS"]$.

**Time Alignment.** We align the Wav2Vec2-generated tokens with the BERT tokens using character-level matching. In particular, for each audio frame, we assign the aligned BERT embedding as its feature. If a match is not found, we fill the gap by using the nearest neighboring non-zero features, ensuring a smooth transition in the feature sequence over time.

## A.3   15D MOTION REPRESENTATION

We refer to NeMF (He et al., 2022) represent the motion at each time step $t$ using a 15-dimensional (15D) feature vector for each joint. This representation captures only local motion. The 15D motion representation, $\mathbf{X}_t \in \mathbb{R}^{J \times 15}$, includes:

Firstly, the joint positions $\mathbf{x}_t^p \in \mathbb{R}^{J \times 3}$ represent the 3D coordinates of each joint relative to the root joint, providing the skeletal pose at time $t$. Secondly, the joint velocities $\dot{\mathbf{x}}_t^p \in \mathbb{R}^{J \times 3}$ capture the rate of change of joint positions.

Additionally, the representation includes joint rotations $\mathbf{x}_t^r \in \mathbb{R}^{J \times 6}$, which encode the orientation of each joint in a 6D rotation format (Zhou et al., 2019), allowing for a more robust and unambiguous rotation representation. Lastly, the angular velocities $\dot{\mathbf{x}}_t^r \in \mathbb{R}^{J \times 3}$ describe the rotational speed of each joint. This unified representation enables a detailed and comprehensive modeling of both the position and movement dynamics of each joint.

## A.4 LIMITATIONS

The ACInterp inputs 2D pose images from linear 2D pose blending. During inference, the image-level pose guidance is obtained by linearly blending detected 2D pose sequences. Since the linear blending is applied independently to each axis, the result of potential 3D blending $(x_1, y_1, z_1) \to (x_2, y_2, z_2)$ is equivalent to blending 2D $(x_1, y_1) \to (x_2, y_2)$ for the $x$- and $y$-axes. However, when the GT motion between 8-interpolated frame is non-linear, the generated results is slightly differ from the GT. We calculate the linear blending could work, *i.e.*, with a 2D pose error smaller than threshold 0.005, on 83% clips for Talkshow-Oliver Dataset.

Besides, our method requires reference videos with low-dynamic backgrounds, such as TalkShow Oliver and YouTubeTalk datasets. Our methods do not work well on speakers with high-dynamic backgrounds. For example, other speakers in TalkShow often interact with the background black-board or move around while talking, resulting in non-stable backgrounds. The blending of highly different backgrounds within very short durations, *e.g.*, half a second, makes the results unnatural.

## A.5 TRAINING DETAILS AND SETTINGS

We trained AuMoCLIP with a learning rate of $5 \times 10^{-4}$, a batch size of 64, on a single Nvidia L4 (24 GB) GPU for 30 hours. For ACInterp we leveraged the pre-trained weights from the reproduced AnimateAnyone by Moore-Thread, then it was finetuned with a learning rate of $1 \times 10^{-5}$, a batch size of 16 for the image stage, and a batch size of 4 for the video stage, using 4 A100 (80 GB) GPUs. The image and video stages were trained for 30k and 20k iterations, respectively, requiring 5-6 days.

## A.6 DETAILS OF MERGING SMALLER SCCS TO THE LARGEST SCC

The graph pruning methodology enhances the connectivity of the motion graph $G$ by merging its strongly connected components (SCCs). We firstly decompose the graph $G$ into strongly connected components (SCCs), which are maximal subgraphs where every node is reachable from every other node within the same subgraph, denoted as $G_{\text{SCC}} = \{G_0, G_1, \ldots, G_n\}$, where $|G_k|$ represents the size (number of nodes) of the $k$-th SCC. Then, we select the largest SCC $G_m$ as the primary component for merging.

Each smaller SCC $G_i$ ($G_i \neq G_m$) is analyzed to determine whether any of its nodes are not in $G_m$. We will try to merge the $G_i$ to $G_m$ If 1) disconnected nodes are found and there is more than 30 disconnected nodes (1-second video), and 2) if the number of nodes in an SCC is smaller than $n$ (set to $n = 100$ in our implementation). These rules are to prevent merging small and isolated nodes into the main SCC.

Then, for each node $u$ in $G_m$ and each node $v$ in $G_i$, we compute the distance $d(u, v)$, where the distance is the similarity for pose positions on 3D space and IOU distance on 2D space. We found the closest pair of nodes $(u, v)$ that minimizes the distance. After determining the closest pair, we add bi-directional edges $e_{u,v}$ and $e_{v,u}$ to the original graph $G$, for effectively merging $G_i$ into $G_m$. Finally, this iterative process produces an enhanced graph $G'$ where paths of any desired length could be sampled from any starting node.

