# OpenReview forum: "TANGO: Co-Speech Gesture Video Reenactment with Hierarchical Audio Motion Embedding and Diffusion Interpolation"
_ICLR.cc/2025/Conference — ICLR 2025 Oral_

### Official Review · Reviewer_JX53 · 2024-10-15

**Soundness:** 4
**Presentation:** 4
**Contribution:** 4
**Rating:** 8
**Confidence:** 5

**Summary:**

The paper presents TANGO, a novel framework for generating co-speech body-gesture videos that effectively synchronizes gestures with speech audio. The key contributions of TANGO include:

Motion Graph-Based Retrieval: TANGO enhanced the motion graph-based model with more fine-grained designed AuMoCLIP to obtain relevant body gestures that align with the audio input, enhancing the realism and relevance of the generated gestures.

Diffusion-Based Interpolation Network: The framework utilizes a diffusion-based approach for video frame interpolation, which improves the quality and fluidity of the generated videos, addressing limitations found in previous methods.

The authors conduct extensive evaluations, including both objective metrics and user studies, demonstrating that TANGO outperforms existing methods like Gesture Video Reenactment (GVR) and Speech-Driven Templates (SDT) in terms of video quality, audio-motion alignment, and overall user preference.

**Strengths:**

For the method design:

1. This work presents a retrieval framework for gesture generation. It resolved a significant issue by the former work GVR (retrieval naively relies on audio onset features and keyword matching)

2. The author represent the gesture patterns within the video as graph and present an efficient pruning strategy to achieve long-sequence representation. This strategy seems effective.

3. The author propose the AuMoCLIP for aligning the two modalities and demonstrate an effective merging of text semantics, high-level audio, and low-level audio features. The ablation study for low-level and high-level retrieval is insightful with very detailed analysis and design. The author analyzed how the temporal alignment of two modalities can be achieved and also the high level semantics.

4. While uplifting the geature pattern generation to the pixel-level video synthesis, the author did not simply reply on commonly used ReferenceNet-like modules but in addition propose homography background offset flow for the consideration of camera changes for realistic videos. The author did very deep analysis of temporal dependencies and resolved the background and jittering problems introduced by camera motion commonly seen in video generations.


For potential insight and benefits for Future studies:
1. Auto-Evaluation: The AuMoCLIP seems to be a very good generalizable evaluation method for automatic evaluation of gesture motion naturalness conditioned on speech. Unlike FGD, diversity or Beat Alignment (BAS), which cannot precisely include the naturalness of motion patterns, the semantics or beat alignment of audio and speech, this model might be very beneficial for future auto-evaluations, though the author did not propose this point.

2. Audio and Gesture Motion Modality alignment: Previous works in this domain have very limited analysis of the how the speech signal functions as the trigger for gesture patterns. Many previous works, (like DiffSheg, CaMN, TalkShow, EMAGE, AMUSE, etc) only designed the generator by incorporating the audio features or text as the control signals to drive the gesture generation but never analyzed how the inner relationships of different modality should be. GestureDiffuCLIP did preliminary studies in aligning the speech word context and gesture motion patterns but ignored the temporal alignment between modalities. In this work, the author did very detailed analysis of the low-level retrieval and high-level retrieval, reconsider the temporal alignment and semantic alignment (low-level indicates the temporal alignment, I consider the local temporal contrastive loss might functions for the beat-dynamic of the gestures, high-level indicates the global semantic alignment of the two modalities. The ablation of the retrieval is very insightful)

**Weaknesses:**

1. The dataset for comparison are limited. The author only selected Oliver in the Show dataset and a small scale in-the-wild YouTube videos. (However, as far as I know, this filed lacks of high-quality data, PATS is another option, but it contains too many low-quality/low-resolution videos and blurry frames for large shoulder and hand motions)

2. It seems like the temporal contrastive learning is based on the low-level features obtained from the CNN encoder. I am not quite sure if I understand it correctly. It will be better if the author can include this information in the final draft for easy understanding.

3. For the method design, in case I am uncertain of some technical details, I will present some potential weakness in the Question section instead of here.

4. For the gesture motion representation, the author utilized 3D SMPL/SMPL-X template model parameters. I assume the author based on the first frame of the given image, do the tracking to obtain the camera and based on the camera to project the generated 3D joints onto the 2D-space for image-level pose guided generation. I think the author could add this information to the final draft.

5. Some of the training details of the model is missing, like what are the GPUs for training, time for training, learning rate.

**Questions:**

1. Representing the gesture as the motion graph: While it is good to represent the gesture patterns from the video as the graph, the retrived features are discrete tokens not uniformly sampled. Will this reduce the quality of gesture generation as the speech audio in natural should be continuous?

2. It seems like for high-level representation, the author only used one transformer layer upon the low-level features from CNN encoders. Is it possible the limited number of transformer level leads to a low performance for the global retrieval? I have this doubt because the author draw inspiration from Wav2Vec 2 for the model design. However, Wav2Vec ustilized 11 transformer layers upon the CNN encoder and the self-supervised learning is also based on the transformer layer feature. Is it possible adding additional layers of transformers can help improve the high-level retrieval?

3. For one technical detail of the paper, why random search of low-level retrieval has a 25% accuracy? In addition, can the author include some other metrics like F1 score, recall? There should be mostly negative samples for the retrieval if my understanding is correct.

4. One further question based on weakness 3, why the temporal contrastive learning is used for the low-level feature instead of applied to high-level features from transformer layers?

5. Refer to Weakness 5, I am assuming the the camera during inference will be kept the same. One claim the author mentioned is "homography background offset flow to eliminate artifacts due to camera parameter changes". Does this only apply for the training but not inference for stablizing training?

---

> ### Author Response · Authors · 2024-11-24
> **Response to Reviewer JX53 (1/2)**
>
> Dear reviewer,
>
> thank you for your time and comments! We list our responses to clarify some misunderstandings in the weaknesses and answer your questions below. If you have any remaining questions or concerns, we are happy to discuss and address them.
>
> **W1: Why only select speaker Oliver in TalkShow, and could the authors provide more?**
>
> Our method requires reference videos with low-dynamic backgrounds, such as TalkShow Oliver and YouTubeTalk datasets as mentioned in Appendix A.1. Therefore, we didn’t compare other speakers with high-dynamic backgrounds. Specifically:
>
> 1. We add the discussion for this limitation in Appendix A.4, explaining why our methods do not work well on speakers with high-dynamic backgrounds. Other speakers in TalkShow often interact with the background blackboard or move around while talking, resulting in non-stable backgrounds. The blending of highly different backgrounds within very short durations, e.g., half a second, makes the results unnatural.
>
> 2. Experiments on Oliver and YouTubeTalk could demonstrate that, in low-dynamic cases, our results outperform previous works. Since YouTubeTalk includes 13 speakers (compared to TalkShow's 4), our experiments on 12 YouTubeTalk speakers are zero-shot, covering both in-domain and out-of-domain comparisons.
>
> **W2: Details of where to obtain the low-level features**
>
> We have redesigned Figure 3 for clarification. Low-level features are from two CNN encoders:
> 1. Fixed pretrained features from the CNN part of Wav2Vec2.
> 2. Learned features from our CNN encoder, which uses Wav2Vec2 CNN encoder architecture.
>
> **W4: Details of how to get image-level pose guidance**
>
> During inference, the image-level pose guidance is obtained by linearly blending detected 2D pose sequences. Since the linear blending is applied independently to each axis, the result of 3D blending $\((x_1, y_1, z_1) \ \rightarrow (x_2, y_2, z_2)\)$ is equivalent to blending 2D $\((x_1, y_1) \ \rightarrow (x_2, y_2)\)$ for the $x$- and $y$-axes. We use DWPose for 2D pose detection and have added this explanation to Appendix A.4.
>
> **W5: Training Details and Settings**
>
> We added these details to Appendix A.5 and will release the training configs. We trained AuMoCLIP with a learning rate of $5 \times 10^{-4}$, a batch size of 64, on a single Nvidia L4 GPU for ~30 hours. ACInterp was trained with a learning rate of $1 \times 10^{-5}$, a batch size of 16 for the image stage, and a batch size of 4 for the video stage, using 4 A100 (80 GB) GPUs. The image and video stages were trained for 30k and 20k iterations, respectively, requiring around 6-7 days.

---

> ### Author Response · Authors · 2024-11-24
> **Response to Reviewer JX53 (2/2)**
>
> **Q1: Continuity for the gestures on the motion graph**
>
> Each node represents a single-frame gesture. Thus, the sample rate matches the FPS of the original video, ensuring smooth transitions in the retrieved video. We fixed this explanation in Section 3.1
>
> **Q2: Is it possible that adding additional layers of transformers can help improve high-level retrieval?**
>
> Yes, adding layers can improve high-level performance from $19.54$% to $20.39$% when increasing from 1-layer to a 12-layer transformer. However, this architecture requires 35 GB of GPU memory, exceeding the 24 GB available on an Nvidia L4. Choosing a 1-layer transformer with 22.5 GB memory makes training feasible on L4 GPUs.
>
> **Q3: Details of low-level retrieval evaluation**
>
> For low-level retrieval, we added a margin $\pm2$ frames to define positive samples. Given a retrieval candidate pool of 16 frames, frames 7-10 are considered rhythm matches, while the remaining frames are not matches. A random search achieves a 4/16 ($25$%) accuracy. Frames outside the 16-frame range (half a second) were not considered, as low-level features focus on matching the audio and gesture rhythm precisely within this interval.
>
> **Q4 (W3): Why use CNN, not transformers, to encode low-level features**
>
> It is unnecessary to use additional transformer layers for low-level features. Our motivation is to limit the perceptual field of the network using shallower, non-global architectures, focusing only on local rhythm. Since the local match between speech and gesture can theoretically be decided locally. Deeper networks and additional transformers perform similarly with shallow CNN here but converge more slowly.
>
> **Q5: Where to apply the homography background offset**
>
> Yes, Homography background offsets are set to zero during inference and applied only during training.

---

> > ### Comment · Reviewer_JX53 · 2024-11-24
> >
> > Thank you to the authors for the detailed responses, which addressed my concerns and questions thoroughly. Overall, this is a solid and insightful paper that holds significant value for advancing research in this domain. I maintain my score of 8. While ICLR does not provide the option to score a paper as 9, I consider this paper to be close to top-tier quality. I did not give a 10 because I am uncertain about the criteria for what defines a "perfect" paper.
> >
> > I encourage the AC and PC to carefully consider the value of this work during their discussions and to combine other reviewers’ opinions to reach a fair and objective evaluation. I strongly believe this paper deserves attention from the community and will have a positive impact on future research.

---

> > > ### Author Response · Authors · 2024-11-25
> > > **Thanks for Your Comment**
> > >
> > > Thank you for the positive comments! Your comments and suggestions help us improve the paper further, and your recognition of our work has greatly encouraged all the authors!

---

### Official Review · Reviewer_UxDj · 2024-10-20

**Soundness:** 4
**Presentation:** 4
**Contribution:** 4
**Rating:** 8
**Confidence:** 5

**Summary:**

This paper proposes a framework for generating co-speech body-gesture videos, dubbed TANGO. TANGO integrates a specifically designed Gesture Video Reenactment (GVR) module, which facilitates the realistic portrayal of gestures, alongside a hierarchical joint embedding space known as AuMoCLIP, which enhances the quality of video generation. The authors conducted extensive experiments that demonstrate the method's superiority over existing approaches in terms of well-rounded metrics and visualization results.

**Strengths:**

This paper proposes a framework for generating co-speech body-gesture videos, dubbed TANGO. TANGO integrates a specifically designed Gesture Video Reenactment (GVR) module and a hierarchical joint embedding space known as AuMoCLIP to facilitate high-quality video generation. The strengths of this work are summarized as follows:
1. The authors present a novel and robust technical contribution through the introduction of a graph network baseline that adaptively incorporates the GVR and AuMoCLIP modules.
2. The motivation for this research is clearly articulated, and the insights provided are compelling. It is important to highlight that a unified framework for co-speech gesture modeling from audio to video sequences holds significant relevance in this community.
3. The dataset introduced in this paper is designed to facilitate ongoing research in gesture generation, providing valuable resources for future studies.
4. The visualization results are well-executed, allowing reviewers to thoroughly evaluate the findings and methodologies presented.

**Weaknesses:**

However, there are still some questions for me. I encourage the authors to give reasonable responses. The final rating would be raised if the solution can solve my concerns.
1. Some important related works might be missed, and I encourage the authors to discuss with them. For example, both of these two methods leverage the audio signal as input to generate human postures.

[1] Qi, X., Pan, J., Li, P., Yuan, R., Chi, X., Li, M., ... & Guo, Y. (2024). Weakly-Supervised Emotion Transition Learning for Diverse 3D Co-speech Gesture Generation. In Proceedings of the IEEE/CVF Conference on Computer Vision and Pattern Recognition (pp. 10424-10434).

[2] Mughal, M. H., Dabral, R., Habibie, I., Donatelli, L., Habermann, M., & Theobalt, C. (2024). ConvoFusion: Multi-Modal Conversational Diffusion for Co-Speech Gesture Synthesis. In Proceedings of the IEEE/CVF Conference on Computer Vision and Pattern Recognition (pp. 1388-1398).

2. In the experimental setting, it is very interesting to explore the frame interval with different settings rather than 4. Please note that this is not an ask for additional experiments in the rebuttal, and this does not impact the paper score. I encourage authors to conduct this discussion in the future.

3. In the user study, did the authors assess participant engagement, specifically regarding whether responses were submitted too quickly or too slowly, and whether any such responses should be discounted? For example, whether the authors considered using response time thresholds or including control questions to check for consistent responses. I encourage authors to add these discussions in the appendix.

**Questions:**

Please refer to Weaknesses.

---

> ### Author Response · Authors · 2024-11-24
> **Response to Reviewer UxDj**
>
> Dear reviewer,
>
> thank you for your time and comments! We list our responses to answer your questions below. If you have any remaining questions or concerns, we are happy to discuss and address them.
>
> **Q1: Discussion about two additional related works.**
>
> Thanks for the suggestion, we add the discussion of [Qi et al. 2024] and [Mughal et al. 2024] in the related work: The performance of the audio2pose stage has been improved by emotion-aware architecture [Qi et al. 2024] and diffusion models [Mughal et al. 2024].
>
> **Q2: Low-level retrieval training setting.**
>
> Thanks and we report performance variance based on the low-level frame margin value. The best parameter (frame margin) of the low-level retrieval was obtained by grid search:
>
> | **Margin Value** | **1-frame**  | **2-frame**   | **3-frame**   | **4-frame**   | **6-frame**   | **8-frame**   |
> |------------------|--------|---------|---------|---------|---------|---------|
> | **Low-Level Performance (%)** | 51.19% | 59.03% | 64.17% | **65.57%** | 57.83% | 44.80% |
>
>
> **Q3: Details of user study.**
>
> For the user study, we employed a two-step process to filter the participants. First, we conducted a quick preliminary test with 3 test samples (9 questions) and initially invited 60 participants. We filtered out participants who: (1) consistently selected the same answer across all questions (e.g., always choosing option C), or (2) demonstrated an inability to understand the task descriptions, such as selecting a blurry video as having high Video Texture Quality. After this initial filtering, we retained 47 participants who passed the criteria to participate in the main user study. Following the completion of the main study, we double-checked the responses to ensure that participants did not consistently select the same answer again, further ensuring the quality of their responses. We did not record the time participants took to complete the form this time, but it is worth considering these rules and time-tracking for future user studies.

---

> ### Comment · Reviewer_UxDj · 2024-11-25
> **Response to authors**
>
> Dear authors,
>
> Thank you very much for your efforts to address my concerns.
>
> I have read all the reviewers' comments and I think this is a novel and valuable work. Since I have already given it an 8 score, I will maintain the current score. I suggest accepting this paper in ICLR, sincerely.
>
> Best
>
> Reviewer  UxDj

---

> > ### Author Response · Authors · 2024-11-27
> > **Thanks for Your Comment**
> >
> > Thank you for your positive feedback! Your recognition of our work has greatly motivated us, and your suggestions during the review helped us enhance the quality of our paper.

---

### Official Review · Reviewer_NVti · 2024-11-02

**Soundness:** 3
**Presentation:** 2
**Contribution:** 3
**Rating:** 8
**Confidence:** 4

**Summary:**

The authors present a co-speech gesture generation approach through a gesture video reenactment framework. To align video segments with audio, the author proposes an audio-motion joint embedding space using contrastive learning.

Furthermore, an existing diffusion-based, pose-driven video generation model is employed to produce interpolated frames. By incorporating an additional reference motion module and a homography-based background flow, the method achieves improved appearance consistency.

Both qualitative and quantitative comparisons with existing state-of-the-art methods demonstrate that the proposed method outperforms alternatives.

**Strengths:**

1. Compared to existing works, the video results of the proposed method are strong in terms of both visual quality and audio-gesture alignment.
2.  The authors conduct thorough experiments to validate the effectiveness of the proposed method.
3. In terms of novelty, I would categorize this work as moderate. Although using contrastive learning for audio-motion retrieval is not new, the author deserves credit for applying it within the speech gesture retrieval domain. Beyond this, other contributions seem more incremental, such as adding graph pruning to GVR and incorporating an additional reference motion module and homography-based background flow to the animation process.

**Weaknesses:**

1. Some claims are not sufficiently justified. For example:

    The author claims, “BERT captures high-dimensional language semantics rather than just ‘audio textures,’ which is critical for the co-speech gesture retrieval task.” However, there is no analysis showing that the addition of BERT features actually generates language-semantics-aware gestures. It’s possible that the performance improvement is due to alignment information alone. If the author could conduct an experiment comparing the use of BERT features with using only CTC alignment information, this might demonstrate the necessity of BERT. Additionally, if the author could provide video examples showing that the generated gestures become more language-semantics-aware with the addition of BERT, it would further support the claim.

2. Since the authors compare their work to other co-speech gesture video generation methods, it would be beneficial to include recent state-of-the-art works, such as *Co-Speech Gesture Video Generation via Motion-Decoupled Diffusion Model (S2GMM).*”If the authors could add this to their evaluation, it would make the proposed work's claim to be state-of-the-art more convincing.

3. Both the objective and subjective evaluations use only one character, Oliver, making it difficult to verify whether the proposed approach would outperform other methods in different circumstances.The authors could perform a comparison experiment with an additional character to demonstrate the generalizability of their approach. Furthermore, if the proposed method does not perform well on characters commonly used by other state-of-the-art methods (e.g., TALKSHOW, PATS), it would be essential to discuss these potential limitations.

4. Writing and formatting improvements are needed for easier comprehension. For example:

- 4.1 Abbreviations:

    - GVR stands for Gesture Video Reenactment in the abstract, but GVR is later used to refer to the specific work (Zhou et al., 2022).

    - The abbreviation for (Liu et al., 2022c) should be ANGIE, not ANGLE.

- 4.2 Tables:

    - In Table 2, methods are listed as columns, while in Table 1, they are listed as rows.

    - There are no captions to explain some non-trivial numbers in the tables, which could be helpful for interpretation.


5. Some explanations and implementation details are unclear or missing. Please refer to the question section for further clarification.

**Questions:**

1. The appendix mentions, “We evaluate our approach using multiple few-shot sets.” Are each of the proposed models (AuMoCLIP, ACInterp) trained separately on only one of these few-shot sets?

2. The author states, “we leverage the power of the two-stage (pose2image and image2video) video generation diffusion model, AnimateAnyone.” However, there are no details on what "leverage" means in this context. Are they merely using the model’s structural concept, or are they initializing with a pre-trained version of AnimateAnyone?

3. Why does Section 4.4 use a different test set for evaluation?

4. Why are there missing results in Table 3?

5. In Figure 5, why is there a white segment within the low-level and high-level feature representations? According to the paper, it should include only the Wav2vec CNN feature and the Wav2Vec transformer feature with the BERT feature.

**Details Of Ethics Concerns:**

The author releases a small-scale YouTube video dataset with limited information about the videos, making it difficult to address potential ethical concerns associated with the dataset.

---

> ### Author Response · Authors · 2024-11-24
> **Response to Reviewer NVti (1/2)**
>
> Dear reviewer,
>
> thank you for your time and comments! We list our responses to answer your questions and concerns in weaknesses below. If you have any remaining questions or concerns, we are happy to discuss and address them.
>
> **W1: An experiment comparing the use of BERT features with only CTC alignment information**
>
> We have conducted an additional experiment replacing the BERT word embedding with a randomly initialized word embedding. This retains the CTC alignment information but removes the clustering in the word semantic space.  It performs a score of  $14.28$%, which is higher than without CTC $12.73$\% but lower than BEAT embedding $15.68$\%. Demonstrate BERT contribute to high-level embedding by both CTC alignment information and word clustering information.
>
> **W2: Comparison with S2GMM**
>
> We have added objective comparisons with S2GMM in Table 1 to demonstrate our state-of-the-art performance.
> | **Method**                     | **FVD ↓** | **FGD ↓** | **BC ↑** | **Diversity ↑** | **FVD ↓** | **FGD ↓** | **BC ↑** | **Diversity ↑** |
> |--------------------------------|-----------|-----------|----------|-----------------|-----------|-----------|----------|-----------------|
> | S2G-Diffusion (He et al., 2024a) | 2.007     | 4.799     | 0.393    | 3.398           | 5.835     | 5.011     | 0.439    | 2.625           |
>
>
>  **W3: Why only select speaker Oliver in TalkShow, and discuss the possible limitations for other speakers.**
>
> Our method requires reference videos with low-dynamic backgrounds, such as TalkShow Oliver and YouTubeTalk datasets as mentioned in Appendix A.1. Therefore, we didn’t compare other speakers with high-dynamic backgrounds. Specifically:
>
> 1. We added discussion for this limitation in Appendix A.4, explaining why our methods do not work well on speakers with high-dynamic backgrounds. Other speakers in TalkShow often interact with the background blackboard or move around while talking, resulting in non-stable backgrounds. The blending of highly different backgrounds within very short durations, e.g., half a second, makes the results unnatural.
>
> 2. Experiments on Oliver and YouTubeTalk could demonstrate that, in low-dynamic cases, our results outperform previous works. Since YouTubeTalk includes 13 speakers (compared to TalkShow's 4), our experiments on 12 YouTubeTalk speakers are zero-shot, covering both in-domain and out-of-domain comparisons.
>
> **W4: Writing and formatting improvements**
>
> Thank you for pointing this out. Gesture Video Reenactment (GVR) and \((Zhou et al., 2022)\) refer to specific works, with full names, abbreviations, and author-year formatting. We have corrected the name for ANGIE.

---

> > ### Comment · Reviewer_NVti · 2024-11-26
> >
> > Dear Author,
> >
> > Thank you for conducting additional experiments to address my concerns.
> > This highlights the strength and thoroughness of your work, and I sincerely appreciate the effort. I’ve updated my score from 6 to 8 to better reflect my evaluation.
> >
> > Best,
> >
> > Reviewer NVti

---

> ### Author Response · Authors · 2024-11-24
> **Response to Reviewer NVti (2/2)**
>
> **Q1: Are AuMoCLIP, and ACInterp trained separately on the TalkShow-Oliver dataset.**
>
> No, our proposed models are trained in one time on all TalkShow-Oliver training sets. Only the training of the Gesture Video Reenactment [Zhou et al. 2022] model is separated since it requires training the network on each reference video.
>
> **Q2: Using the model’s structural concept, or initializing with a pre-trained version of AnimateAnyone?**
>
> Yes, both, we leveraged the pre-trained weights from the reproduced AnimateAnyone by Moore-Thread. We added these details in Appendix A.5.
>
> **Q3: Why does Section 4.4 use a different test set for evaluation?**
>
> The test set in Section 4.4 is the same as in other sections, derived from the same videos. Since the test videos in Sections 4.2 and 4.3 vary in length (e.g., 3 to 10 seconds). For Section 4.4, we evenly sampled 8-frame clips, resulting in a 368-video test set for evaluating blending. We revised Section 4.4 to clarify this.
>
> **Q4: Why are there missing results in Table 3?**
>
> We focus on highlighting the effects of the proposed features. Reporting performance on levels where the models were not explicitly trained is unnecessary as the scores are poor. For example:
>
> - **(Rows 3 to 4):** Keyword matching is designed for high-level retrieval, so we only report high-level scores. Its low-level performance is $24.10$\%, close to random search $25.00$\%. Similarly, onset matching achieves $0.873$\% for high-level retrieval but is intended for low-level tasks.
>
> - **(Rows 5 to 9):** These rows optimize high-level similarity using the CLS token without explicitly optimizing low-level similarity. Using features from each frame for low-level retrieval results in scores of $30.58$\%, $29.12$\%, $28.96$\%, $29.66$\), and $30.13$\% for Rows 5 to 9, all lower than the baseline in Row 10 ($47.94$\%).
>
> - **(Row 11):** Training for low-level similarity does not explicitly consider high-level retrieval, resulting in a high-level score of $3.71$\%, which is lower than the high-level baseline $12.73$\%.
>
> **Q5: What is the white segment In Figure 5.**
>
> The white segment represents a learned feature combined with the fixed pre-trained features (Wav2Vec CNN, Wav2Vec transformer, and BERT features). Using only the pre-trained features achieves $3.79$\% for high-level retrieval, while adding the learned feature significantly improves performance to $11.84$\%, which is the key to make the pipeline work. We have redesigned and updated Figure 5 for clarity.

---

> ### Author Response · Authors · 2024-11-27
> **Thanks for Your Comment**
>
> Thank you for your thoughtful comments and updating the score! Your acknowledgement of our efforts has been deeply encouraging to all authors and guided us in refining the paper!

---

### Official Review · Reviewer_8mqD · 2024-11-04

**Soundness:** 4
**Presentation:** 4
**Contribution:** 4
**Rating:** 10
**Confidence:** 5

**Summary:**

The paper introduces TANGO, a framework for generating high-fidelity co-speech gesture videos using a motion graph-based retrieval approach. It addresses audio-motion misalignment and visual artifacts in previous methods by implementing two key innovations: the AuMoCLIP, a hierarchical audio-motion joint embedding space for improved cross-modal alignment, and ACInterp, a diffusion-based interpolation network for generating high-quality transition frames. TANGO effectively synchronizes body gestures with speech audio, outperforming existing methods on datasets like Talkshow-Oliver and YouTube Business. The framework represents a significant advancement in gesture video generation, providing an open-source solution that integrates these novel components.

**Strengths:**

1. Contributions to Retrieval, Interpolation, and Datasets: The paper introduces TANGO, which enhances co-speech gesture video generation through improved retrieval methods, a diffusion-based interpolation model, and the introduction of the YouTube Business dataset.
2. Contribution to Open Source Gesture Generation: By making their code, pretrained models, and datasets publicly available, the authors significantly advance the field of gesture generation and encourage collaboration among researchers.
3. Comprehensive Experiments: The extensive evaluation of TANGO using various quantitative metrics and user studies demonstrates its superiority over existing state-of-the-art methods in generating realistic and audio-synchronized gesture videos.

**Weaknesses:**

1. Clarification on Graph Pruning Methodology: The paper does not provide sufficient detail on how the strongly connected component (SCC) subgraphs are merged, which is a critical operation in the Graph Pruning section.
2. Inconsistencies in Visual Representation: In Figure 3, the use of blue and green to represent different video motion clips lacks rigorous color correspondence during transitions, potentially leading to misunderstandings; similarly, Figure 5 uses yellow for both the Wav2Vec2 transformer feature and input audio, which could cause confusion.
5. Typographical Errors: There is a typographical error in Figure 4 where "merge" should be corrected to "merging."
4. Testing Results: The generated gesture movements exhibit stuttering, which undermines the fluidity and realism of the videos.

**Questions:**

See weaknesses above.

**Details Of Ethics Concerns:**

There are copyright concerns regarding the proposed dataset, YouTube Business.

---

> ### Author Response · Authors · 2024-11-24
> **Response to Reviewer 8mqD**
>
> Dear reviewer,
>
> Thank you for your time and comments! We list our responses to answer your questions below. If you have any remaining questions or concerns, we are happy to discuss and address them.
>
> **Q1: Clarification on graph pruning methodology**
>
> We add Appendix A.6 to explain the Details of Merging Smaller SCCs into the Largest SCC:
>
> The graph pruning enhances the connectivity of the motion graph $G$ by merging its strongly connected components (SCCs). We firstly decompose the graph $G$ into strongly connected components (SCCs), which are maximal subgraphs where every node is reachable from every other node within the same subgraph, denoted as $G_{\text{SCC}} = \{G_0, G_1, \dots, G_n\}$, where $|G_k|$ represents the size (number of nodes) of the $k$-th SCC. Then, we select the largest SCC $G_m$ as the primary component for merging.
>
> Each smaller SCC $G_i$ ($G_i \neq G_m$) is analyzed to determine whether any of its nodes are not in $G_m$. We will try to merge the $G_i$ to $G_m$ If 1) disconnected nodes are found and there are more than 30 disconnected nodes (1-second video), and 2) if the number of nodes in an SCC is smaller than $n$ (set to $n = 100$ in our implementation). These rules are to prevent merging small and isolated nodes into the main SCC.
>
> Then, for each node $u$ in $G_m$ and each node $v$ in $G_i$, we compute the distance $d(u, v)$, where the distance is the similarity for pose positions on 3D space and IOU distance on 2D space. We found the closest pair of nodes $(u, v)$ that minimizes the distance. After determining the closest pair, we add bi-directional edges $e_{u,v}$ and $e_{v,u}$ to the original graph $G$, for effectively merging $G_i$ into $G_m$. Finally, this iterative process produces an enhanced graph $G'$ where paths of any desired length could be sampled from any starting node.
>
> **Q2: Color consistency on figures.**
>
> Thanks and we updated Figure 3 and Figure 5. For Figure 3, we change the color during transitions from black to blue and green to enhance the color consistency. For Figure 5 we use gray block to represent the input audio wave and blue for Wav2Vec2 features.
>
> **Q3: Typographical errors**
>
> Thanks and we fixed the merge to merging in Figure 4.
>
> **Q4: The generated gesture movements exhibit stuttering.**
>
> This problem could be reflected by setting a more strict threshold for creating the Graph. There is a trade-off between continuity and diversity and we decided to give this control parameter to users. A higher threshold such as 1.2 will have a high diversity but somewhat jitter results. A low threshold such as 0.8 will have more natural but low diversity results. In our main paper, we set the threshold as 1.0.
>
> We have also included 15 additional subjective video results in the supplementary materials (with this threshold of 1.0). We hope this addition could be helpful in addressing concerns about the results' quality.
>
> **Others**
>
> We specifically thank you for the mention of our efforts on the open-source,  this encouragement and understanding give authors confidence to keep doing better for the community.

---

> ### Comment · Reviewer_8mqD · 2024-12-03
> **Response to Authors**
>
> Dear authors:
>
>
> You have thoroughly addressed all my concerns. After multiple careful reviews, I commend your diligent efforts and am impressed by both the exceptional performance and utility of their work. I therefore raise my score to 10.
>
>
> Best,
>
> Reviewer 8mqD

---

> > ### Author Response · Authors · 2024-12-03
> > **Thanks for Your Comment**
> >
> > Thank you for your very positive comments and updating the score! That has been deeply encouraging to all authors, your comments and encouragement definitely help us for refining the paper better!

---

### Meta-Review · Area_Chair_quZA · 2024-12-19

**Metareview:**

The paper introduces a framework for generating high-fidelity co-speech gesture videos using a motion graph-based retrieval approach. It addresses audio-motion misalignment and visual artifacts by introducing two key innovations: a hierarchical audio-motion joint embedding space for improved cross-modal alignment, and a diffusion-based interpolation network for generating high-quality transition frames. All the reviewers gave very positive feedback on this paper. AC agrees with the recommendation of the reviewers and decided to accept this work.

**Additional Comments On Reviewer Discussion:**

All the reviewers gave positive feedbacks and provided high ratings.

---

### Decision · Program_Chairs · 2025-01-22

Accept (Oral)